# Bak and Bcl-xL Participate in Regulating Sensitivity of Solid Tumor Derived Cell Lines to Mcl-1 Inhibitors

**DOI:** 10.3390/cancers14010181

**Published:** 2021-12-30

**Authors:** Viacheslav V. Senichkin, Nikolay V. Pervushin, Alexey V. Zamaraev, Elena V. Sazonova, Anton P. Zuev, Alena Y. Streletskaia, Tatiana A. Prikazchikova, Timofei S. Zatsepin, Olga V. Kovaleva, Elena M. Tchevkina, Boris Zhivotovsky, Gelina S. Kopeina

**Affiliations:** 1Faculty of Medicine, MV Lomonosov Moscow State University, 119991 Moscow, Russia; slsenichkin@gmail.com (V.V.S.); rhododendron.nick@mail.ru (N.V.P.); a-zamaraev@yandex.ru (A.V.Z.); sazonova_82@mail.ru (E.V.S.); anzuev98@gmail.com (A.P.Z.); as213@rice.edu (A.Y.S.); 2Skolkovo Institute of Science and Technology, 121205 Skolkovo, Russia; Tatiana.Alex.Prikazchikova@gmail.com (T.A.P.); t.zatsepin@skoltech.ru (T.S.Z.); 3Faculty of Chemistry, MV Lomonosov Moscow State University, 119991 Moscow, Russia; 4NN Blokhin Russian Cancer Research Center, Department of Oncogenes Regulation, 115478 Moscow, Russia; ovkovaleva@gmail.com (O.V.K.); tchevkina@mail.ru (E.M.T.); 5Institute of Environmental Medicine, Karolinska Institutet, 17177 Stockholm, Sweden

**Keywords:** Bcl-2 family proteins, BH3-mimetics, Mcl-1, cancer therapy, sensitivity

## Abstract

**Simple Summary:**

Apoptosis is one of the best-known types of programmed cell death. This process is regulated by a number of genes and proteins, among which the Bcl-2 protein family plays a key role. This family includes anti- and proapoptotic proteins. Cancer cell resistance to apoptosis is commonly associated with overexpression of the antiapoptotic members of Bcl-2 family proteins, in particular, Bcl-2, Bcl-xL, and Mcl-1. Subsequently, these proteins represent perspective targets for anticancer therapy. Here, using an inhibitory approach, we found that Bak and Bcl-xL regulate sensitivity of cancer cells to Mcl-1 inhibition.

**Abstract:**

BH3 mimetics represent a promising tool in cancer treatment. Recently, the drugs targeting the Mcl-1 protein progressed into clinical trials, and numerous studies are focused on the investigation of their activity in various preclinical models. We investigated two BH3 mimetics to Mcl-1, A1210477 and S63845, and found their different efficacies in on-target doses, despite the fact that both agents interacted with the target. Thus, S63845 induced apoptosis more effectively through a Bak-dependent mechanism. There was an increase in the level of Bcl-xL protein in cells with acquired resistance to Mcl-1 inhibition. Cell lines sensitive to S63845 demonstrated low expression of Bcl-xL. Tumor tissues from patients with lung adenocarcinoma were characterized by decreased Bcl-xL and increased Bak levels of both mRNA and proteins. Concomitant inhibition of Bcl-xL and Mcl-1 demonstrated dramatic cytotoxicity in six of seven studied cell lines. We proposed that co-targeting Bcl-xL and Mcl-1 might lead to a release of Bak, which cannot be neutralized by other anti-apoptotic proteins. Surprisingly, in Bak-knockout cells, inhibition of Mcl-1 and Bcl-xL still resulted in pronounced cell death, arguing against a sole role of Bak in the studied phenomenon. We demonstrate that Bak and Bcl-xL are co-factors for, respectively, sensitivity and resistance to Mcl-1 inhibition.

## 1. Introduction

In multicellular organisms, apoptosis, an evolutionary conserved mechanism of programmed cell death (PCD), tightly controls cell number and tissue homeostasis. Similar to other types of PCD, apoptosis is an important oncosuppressive process that enables the removal of unwanted cells, in particular, cells with damaged genetic material [1]. Resistance to cell death is one of the hallmarks of cancer [2]. This phenomenon underlies not only tumorigenesis, but also resistance to the action of various therapeutic agents.

Members of the B-cell lymphoma 2 (Bcl-2) family are key players in the regulation of the mitochondrial (intrinsic) pathway of apoptosis. This family includes both anti- and pro-apoptotic proteins [3]. Pro-survival (i.e., anti-apoptotic) members, such as Bcl-2, Bcl-xL, Bcl-w, myeloid cell leukemia 1 (Mcl-1), and Bcl-A1, bind and neutralize players of two distinct pro-apoptotic subsets—effector and BH3-only proteins [4]. The effectors of the Bcl-2 family, Bak and Bax, form pores in the outer mitochondrial membrane (OMM) [5], thus leading to the release of various pro-apoptotic factors into the cytosol, which is considered a point of no return in the mitochondrial-mediated apoptotic pathway [6]. In contrast, BH3-only pro-apoptotic proteins do not cause mitochondrial outer membrane permeabilization (MOMP) directly. Instead, these proteins neutralize anti-apoptotic members of the Bcl-2 family and indirectly derepress Bak and Bax. Moreover, some BH3-only proteins (Bim, truncated Bid) can directly activate pore-forming proteins; therefore, they are called “BH3-only activators”. Proteins of another BH3-only subgroup (Puma, Noxa, Bad) do not directly activate Bak and Bax and are called “BH3-only sensitizers” [3,7].

All proteins of the Bcl-2 family contain one or more Bcl-2 homology (BH) domains. As the name suggests, BH3-only proteins have only one such domain, which endows these proteins with pro-apoptotic activity [8]. Both pore-forming and anti-apoptotic proteins of the Bcl-2 family have four BH domains (BH1–BH4), which form a globular structure [3,9]. When activated (e.g., upon binding to BH3-only activators), the effector proteins expose their BH3 domains and switch to a pro-apoptotic state [10]. Anti-apoptotic proteins retain their structures, in which the BH3 domain, together with other BH domains, form a hydrophobic groove. This groove binds the BH3 domains of pro-apoptotic proteins and neutralizes them. Such interactions between pro- and anti-apoptotic proteins underlie the functioning of the Bcl-2 family [6].

An important aspect of the interactions between Bcl-2 family proteins is their different affinities to various partners [3,11]. The pro-apoptotic protein Bim interacts with a wide range of anti-apoptotic proteins (e.g., Bcl-2, Mcl-1, Bcl-xL, Bcl-w, Bcl-A1), while Bad interacts with Bcl-2, Bcl-xL, and Bcl-w, but not with Mcl-1 or Bcl-A1. Noxa, conversely, binds to Mcl-1 and Bcl-A1, but not to other anti-apoptotic proteins [7,12]. The pore-forming protein Bax promiscuously interacts with various anti-apoptotic proteins, while Bak preferentially binds to Bcl-xL and Mcl-1 [11,13,14]. Altogether, these interactions create a complex system of control over MOMP and, therefore, apoptosis induction [3,15].

The ratios of pro- and anti-apoptotic proteins of the Bcl-2 family can determine cell fate between life and death. Disruption of the balance between these two functional groups of proteins leads to the dysregulation of apoptosis, which is often exploited by cancer cells to resist this type of cell death [6]. A number of studies have shown that tumor cells are often characterized by increased levels of anti-apoptotic proteins [16,17,18]. In these cases, neutralization of anti-apoptotic proteins represents an effective strategy for apoptosis induction [19]. For this purpose, BH3 mimetics (small-molecule inhibitors of the pro-survival members of the Bcl-2 family) have been developed [20,21,22]. These compounds bind to the BH3-binding groove of anti-apoptotic proteins and neutralize them. Similar to BH3-only proteins, several BH3 mimetics (e.g., obatoclax and gossypol) interact with various Bcl-2 family proteins. These compounds are widely used in vitro but have not been shown to be effective in clinical trials [23]. BH3 mimetics to individual anti-apoptotic proteins were more successful, and the most important breakthrough was the approval of venetoclax, a selective Bcl-2 inhibitor, for the treatment of patients with chronic lymphocytic leukemia (CLL) and acute myeloid leukemia (AML) (for the latter, in combination with azacitidine, decitabine, or low-dose cytarabine) [24]. In contrast to venetoclax, BH3 mimetics to other Bcl-2 family members have been studied only in preclinical or early phase clinical trials [24]. For a long time, there were no selective and highly active inhibitors of Mcl-1 [23]. However, in the recent five years, a number of BH3 mimetics to Mcl-1, such as A1210477 [25], S63845 [26], AZD-5991 [27], AMG-176 [28], and VU661013 [29], have been described. Several of them have been included in clinical trials for the treatment of patients with hematological diseases, both as single agents or in combination with venetoclax [23]. Hence, studying BH3 mimetics to Mcl-1 is of great importance in the context of their possible use as anticancer agents. In particular, attention should be paid to the predictors of sensitivity and factors of resistance to these agents. Understanding these aspects will facilitate the implementation of Mcl-1 inhibitors in the clinical setting. In this study, we have shown the potential markers for sensitivity of cancer cells to Mcl-1 inhibition and effective combinations of agents to overcome a resistance to BH3 mimetics to Mcl-1.

## 2. Materials and Methods

### 2.1. Cell Lines and Culture

The colorectal carcinoma cell line HCT-116, non-small cell lung carcinoma (NSCLC) cell lines A549 and H23, ovarian carcinoma cell line Caov-4, cervical adenocarcinoma cell line HeLa, neuroblastoma cell lines SK-N-BE(2) and SH-SY5Y were kindly provided by the Department of Toxicology, Karolinska Institutet (Stockholm, Sweden). Cells were cultured in Dulbecco’s Modified Eagle Medium (DMEM) high glucose (Gibco, Waltham, MA, USA) or Roswell Park Memorial Institute 1640 (RPMI-1640) medium (Gibco). The media were supplemented by a mixture of antibiotic-antimycotic penicillin (100 U/mL) plus streptomycin (100 μg/mL) (Gibco), 1 mM sodium pyruvate (PanEco, Moscow, Russia), and 10% fetal bovine serum (Gibco). All manipulations with live eukaryotic cells were performed in a tissue culture hood in sterile conditions. Cells were grown in a CO_2_ incubator (5% CO_2_) at 37 °C and reseeded every two to three days. A measure of 0.15% trypsin solution (Gibco) was used to remove the cells from the plates. Cells at logarithmic growth phase were used for the experiments. Before treatment with BH3 mimetics or cisplatin, the cells were washed with phosphate buffered saline (PBS) solution (PanEco), and fresh medium was added. Cisplatin was purchased from Teva (Petah Tikva, Israel); A-1210477 and S63845 were purchased from Active Biochem (Hong Kong, China), verapamil was from Sigma (Saint Louis, MO, USA)*,* A-1331852 and ABT-199 (venetoclax) were from Selleckchem (Houston, TX, USA).

### 2.2. Western Blot (WB)

Cells were removed from culture dishes using 0.15% trypsin solution or a cell scraper. Next, cells were centrifuged (1000 rcf, 4 min, +4 °C), washed with PBS solution, centrifuged again (1000 rcf, 4 min, +4 °C), and the pellet was lysed in Radioimmunoprecipitation assay buffer (RIPA buffer) containing 50 mM Tris-HCl [pH 7.4], 150 mM NaCl, 2 mM EDTA, 0.5% SDS, 0.5% sodium deoxycholate, 1% NP-40, 1 mM phenylmethylsulfonyl fluoride (PMSF) and cOmplete™ Protease Inhibitor Cocktail (Roche, Basel, Switzerland) for 15 min on ice. After centrifugation (15,000 rcf, 15 min, +4 °C), a part of the supernatant was taken for protein concentration assay, and another part was used for western blot analysis, as previously described [30]. The following primary antibodies were used for western blot: anti- glyceraldehyde 3-phosphate dehydrogenase (GAPDH) (#2118), anti-Bcl-xL (#2764), anti-Bax (#2772), anti-Bim (#2933), anti-Mcl-1 (#5453), anti-Bak (#6947), anti-rabbit full and cleaved caspase-3 (#9662) antibodies (Cell Signaling Technology, Danvers, MA, USA); anti-Bcl-2 (sc-509) (Santa Cruz Biotechnology, Dallas, TX, USA); anti- Poly(ADP Ribose) Polymerase (PARP) (#137653) and anti-tubulin-α (#7291) antibodies (Abcam, Cambridge, UK). HRP-linked goat anti-mouse and anti-rabbit antibodies (#97046 and #97200, respectively; Abcam) were used as secondary antibodies.

### 2.3. siRNA Transfection

Small interfering RNA (siRNA) transfection was performed using RNAiMAX reagent (Thermo Scientific, Waltham, MA, USA) according to the manufacturer’s recommendations. The final concentration of siRNA to Mcl-1 was 100 nM, and time of transfection was 6 h; other siRNAs were used in concentrations 50 nM for 24 h. In experiments with siRNAs, control cells were treated with non-targeting siRNA (Dharmacon, Lafayette, CO, USA) at a final concentration corresponding to that of targeting siRNAs and for the same time periods.

### 2.4. siRNAs

siRNAs were designed and synthesized as previously described [31,32]. In particular, siRNAs targeting Bak, Bax, and Bim were designed to reach the lowest off-target potential, including a minimal overlap with microRNA (miRNA) seed regions to avoid miRNA-like activity and decreased capacity to activate innate immunity [33,34,35]. The generated siRNAs were ranked based on the number of mismatches in the seed region, non-seed region, and cleavage site position. Pyrimidine nucleotides upstream to adenosine residues were replaced with 2′-O-methyl analogues and a phosphorothioate linkage was introduced between two nucleotides at the 3′-end in both siRNA strands to improve nuclease stability. Sets of best scored siRNAs (seven for Bak, five for Bax, and ten for Bim) were synthesized using the phosphoramidite approach, purified by Ion-Exchange High Performance Liquid Chromatography (IE-HPLC), and verified by Liquid chromatography–mass spectrometry (LC-MS), followed by strand annealing, as previously described [36]. For Mcl-1 downregulation, a previously described siRNA was used that has been shown to efficiently silence Mcl-1 (the sense and antisense sequences are, respectively, 5′-GCATCGAACCATTAGCAGAdTdT-3′ and 5′-TCTGCTAATGGTTCGATGCdTdT-3′) [37,38]. Next, siRNA efficiency to downregulate targeted proteins was assessed using western blot. The sequences of siRNAs, as well as the results of testing of their efficiency are shown in the Appendix A, respectively. Two duplexes with the highest efficacy were chosen per each target to avoid off-target effects: siRNAs #1 and #7 for Bak; siRNAs #1 and #5 for Bax; siRNAs #8 and #9 for Bim. The sequences were as follows: Bak siRNAs (gene BAK1)—#1 (sense—5′-GuAcGAAGAuucuucAAAuTsT-3′ and antisense—5′-AUUUGAAGAAUCUUCGuACTsT-3′), #7 (sense—5′-AAGcGAAGucuuuGccuucTsT-3′ and antisense 5′-GAAGGcAAAGACUUCGCUUTsT-3′); Bax siRNA (gene Bax)—#1 (sense—5′-uuuucuGAcGGcAAcuucATsT-3′ and antisense—5′-UGAAGUUGCCGUcAGAAAATsT-3′), #5 (sense—5′-AAcuGAucAGAAccAucAuTsT-3′ and antisense—5′-AUGAUGGUUCUGAUcAGUUTsT-3′); Bim siRNAs (gene BCL2L11)—#8 (sense—5′-AccGAGAAGGuAGAcAAuuTsT-3′ and antisense—5′-AAUUGUCuACCUUCUCGGUTsT-3′), #9 (sense—5′-GuGAccGAGAAGGuAGAcAAuuGcAGcTsT-3′ and antisense—5′-GCUGcAAUUGUCuACCUUCUCGGUcACTsT-3′). Uppercase letters: ribonucleotides, lowercase letters: 2′-O-Methyl nucleotides, s: phosphorothioate.

### 2.5. CRISPR/Cas9 Genome Editing

To generate BAK1 knockout cells, HeLa and H23 cell lines were transfected with pSpCas9-BB-2A-GFP plasmid (GenScript)-containing single guide RNA (sgRNA) coding sequence (GTTGATGTCGTCCCCGATGA) targeting BAK1 gene (GenScript, Piscataway, NJ, USA). Plasmid was used at a final concentration 1 μg/mL. Lipofectamine LTX with Plus Reagent (Thermo Scientific) was utilized for transfection according to the manufacturer’s recommendations. Green fluorescent protein-positive (GFP-positive) cells were selected by cell sorting on the FACSAria III Cell Sorter (BD Biosciences, San Jose, CA, USA). In the case of the H23 cell line, the total population of GFP-positive cells was taken for further experiments since Bak was efficiently knocked-out. However, Bak knockout was inefficient for the total population of HeLa GFP-positive cells. Hence, individual clones of HeLa GFP-positive cells were used to generate populations of cells lacking Bak. To exclude the off-target effects of specific clones, two populations derived from different clones were tested. To exclude the non-specific effects of CRISPR/Cas9 genome editing, Bak-knockout cells were compared to control cells that were transfected with non-targeting control CRISPR/Cas9 plasmid sc-418922 (Santa Cruz Biotechnology, final concentration 1 μg/mL) and sorted similarly to corresponding knockout cells. These cells were designated as “HeLa^control^” and “H23^control^”.

### 2.6. Fluorescence-Activated Cell Sorting Analysis (FACS-Analysis)

Cells were removed from the culture dishes using 0.15% trypsin solution and counted using the Z1 Particle Counter (Beckman Coulter, Chaska, MN, USA). A total of 10^5^ cells were taken for analysis, centrifuged (1000 rcf, 4 min, +4 °C), washed with PBS solution, and centrifuged again (1000 rcf, 4 min, +4 °C). The cell pellet was resuspended in 200 µL Annexin-binding buffer (BD Biosciences). Next, 2 µL Annexin V-FITC (Thermo Scientific) was added, and cells were incubated with Annexin for 15 min in a dark place (at room temperature). Immediately before measurement, propidium iodide (Sigma) was added to the samples to a final concentration 0.5 µg/mL, and the samples were analyzed by the BD FACSCanto II cell analyzer (BD Biosciences).

### 2.7. Sub-G1 Test

After the indicated time of treatment, the cells were collected and fixed in 70% ethanol during 1 h at −20 °C. Then, the cells were washed of ethanol and re-suspended in PBS, supplemented with 1% RNase A and stained with 20 μg/mL propidium iodide for 15 min at 37 °C. After staining, cells were examined using the FACSCanto II cell analyzer.

### 2.8. MTS Assay

The assay is based upon the cleavage of the yellow tetrazolium salt MTS (3-(4,5-dimethylthiazol-2-yl)-5-(3-carboxymethoxyphenyl)-2-(4-sulfophenyl)-2H-tetrazolium) to a water-soluble, orange-colored formazan product which dissolves directly into the culture medium. Briefly, cells were plated into flat-bottomed 96-well plates (Nunc, Denmark) and cultured in DMEM supplemented with 10% FBS overnight. Media was then removed and replaced with 0.1 mL fresh DMEM containing different concentrations of S63845, which ranged from 0.081 to 200 µM. The plates were incubated at 37 °C for 24 h. After treatment, 20 µL of MTS (CellTiter 96 AQueous One Solution Cell Proliferation Assay, Promega, Madison, WI, USA) labeling reagent were added to each well and plates were additionally incubated at 37 °C for 3 h. Following MTS incubation, the spectrophotometric absorbance of the samples was detected by using a microplate reader (Bio-Rad Laboratories, Inc., Hercules, CA, USA) at 480 nm with a reference wavelength of 630 nm. Half-maximal inhibitory concentration (IC50) values were calculated based on log values using GraphPad Prism version 8 software (GraphPad Software, Inc., La Jolla, CA, USA).

### 2.9. TCGA Data Analysis

The mRNA expression data were queried from the cBioPortal in the form of z-score-transformed data based on the Lung Adenocarcinoma Cancer Genome Atlas database (LUAD TCGA; https://portal.gdc.cancer.gov/, accessed on 1 September 2021) and analyzed by R3.6.1 software [39,40]. Z-score mRNA expression relative to normal samples indicated the number of standard deviations from the mean of expression of the same gene in normal samples. The overall survival data of LUAD patients were queried from the cBioPortal and compared between several subsets of patients with high and low expression of selected genes. The “high” and “low” expression of the genes was based on the median of z-score mRNA expression relative to normal samples, above the median and below, respectively (Appendix A). For survival analysis, the 10-year range of observation was selected. The statistical analysis was performed using the log-rank test by R3.6.1 software with the visualization by the Kaplan–Meier plot. To account for multiple comparisons, the Holm–Bonferroni correction was used.

### 2.10. Clinical Sample Collection, Preparation and Processing

Surgically resected specimens were collected from patients with lung adenocarcinomas at the Clinical Oncology Research Institute, N.N. Blokhin Russian Cancer Research Center during the period 2016–2020. After surgical removal, the tumor specimens were frozen and stored in liquid nitrogen. All patients signed informed consent forms according to the legal institutional guidelines and ethical permission. The tumor clinic morphological stages were determined according to the standard tumor TNM classification systems of the International Union Against Cancer (seventh edition). In this study, lung cancer tissues and matched non-cancerous tissues were obtained from the 20 patients. All samples were homogenized using BashingBead Lysis Tubes (Zymo Research, Irvine, CA, USA) and Precellys 24 tissue homogenizer (Bertin Technologies, Montigny-le-Bretonneux, France). A total of 0.5 mL lysis buffer was added per sample (50 mM Tris-HCl (pH 7.4), 150 mM NaCl, 2 mM EDTA, 0.5% SDS, 0.5% sodium deoxycholate, 1% NP-40, 1 mM PMSF, cOmplete™ Protease Inhibitor Cocktail). The whole procedure was carried out in a cold room at +4 °C. Next, samples were incubated for 30 min on ice for cell lysis followed by centrifugation (15,000 rcf, 30 min, +4 °C). Supernatant was taken for western blot analysis. The data of western blot were analyzed by ImageJ 4.1 (LOCI, University of Wisconsin, USA) and R3.6 software. The protein level of each clinical sample was normalized by total protein level based on TGX stain free gel (Bio-Rad, Hercules, CA, USA) and the log2 cancer/normal ratio was calculated.

### 2.11. Data Processing and Statistical Analysis

Western blot images were processed using Image Lab software. Densitometric analysis was performed using Image Lab Software or ImageJ 4.1 (for clinical samples). Flow cytometry data were processed using BD FACSDiva software 7.0 (BD Biosciences). For the statistical analysis, all data were tested for homogeneity of variance and normality using Levene’s and Shapiro–Wilk tests, respectively. For normally distributed data, the Student’s *t*-test was performed to analyze statistically significant differences between groups. *p*-values lower than 0.05 were considered statistically significant.

## 3. Results

### 3.1. S63845 Exhibits Higher Proapoptotic Activity Than A1210477

In this work, we evaluated the activity of two BH3 mimetics to Mcl-1, A1210477 and S63845, which have been previously shown to effectively induce apoptosis in various cell lines [25,26]. Of note, A1210477 was used at the highest concentration of 10 μM, since higher doses of this compound exhibit off-target toxicity due to caspase-dependent, but Bax/Bak-independent cell killing [41] and are also capable of disrupting Bcl-2/Bim complexes [25]. Mcl-1 accumulation is an important pharmacodynamic marker of its inhibition by BH3 mimetics, which has been previously described in numerous studies [25,26,27]. An increase in Mcl-1 levels in response to both tested BH3 mimetics was observed, while the levels of other anti-apoptotic proteins, Bcl-2 or Bcl-xL, remained unchanged (Figure 1A). As previously reported [26], S63845 exhibited higher potency, which can be explained not only due to the differences in affinity to Mcl-1, but also due to the interaction of A1210477 with serum proteins [25,38]. The efficiency of apoptosis induction by both BH3 mimetics was then estimated. For this purpose, the processing of caspase-3 with the formation of catalytically active fragments (p-17/p-19) was evaluated. Additionally, the cleavage of PARP, which is a substrate of effector caspase-3 and -7, was assessed. Lower levels of the full-length PARP and higher levels of the p89 fragment reflect the induction of apoptosis. Remarkably, even at the maximum concentration of 10 μM, A1210477 less efficiently induced apoptosis, as compared to S63845, which was shown in HeLa cells (Figure 1A). At the same time, Caov-4 cells responded poorly to both BH3 mimetics. Hence, this cell line shows less dependence on Mcl-1 and, therefore, requires an additional stimulus for efficient apoptosis induction.

Next, cell death induced by the combination of Mcl-1 inhibitors with the chemotherapeutic agent cisplatin was evaluated. To this end, FACS-analysis with Annexin V-FITC and propidium iodide (PI) staining was used in addition to PARP cleavage assay. Consistent with the data above, S63845 alone was shown to be more effective than A1210477 in HeLa, but not in Caov-4 cells. However, when combined with cisplatin, S63845 demonstrated higher proapoptotic activity in both HeLa and Caov-4 cell lines, as assessed by both PARP cleavage (Figure 1B) and FACS-analysis (Figure 1C). Thus, while both BH3 mimetics interacted with their target, only S63845 exhibited high cell killing efficiency.

The obtained data indicated different proapoptotic activities of A1210477 and S63845, which could be explained by either the limited efficiency of A1210477 in the used concentration range or off-target activity of S63845. To address this issue, we analyzed the proapoptotic activity of the studied compounds in comparison to the specific siRNA-mediated knockdown of MCL1. For this purpose, HeLa cells were treated with A1210477 or S63845, either alone or in combination with cisplatin. Cells without transfection, with non-targeted transfection or transfected with Mcl-1 siRNA were compared. It was shown that A1210477 demonstrated lower antiapoptotic activity compared to the downregulation of Mcl-1 by siRNA, both when these approaches of Mcl-1 inhibition were used alone and when they were combined with cisplatin (Figure 1D,E). In contrast, S63845 demonstrated proapoptotic activity comparable to that of Mcl-1 downregulation (Figure 1D,E). Similar results were also observed in Caov-4 cells, with a difference seen only if BH3 mimetics were combined with cisplatin (Appendix A). Hence, the higher proapoptotic activity of S63845, compared to A1210477, is likely due to the on-target toxicity of this compound, while the efficacy of A1210477 is limited for concentrations which provide its on-target action.

### 3.2. Bak Is Required for Efficient Cell Killing in Response to BH3 Mimetics to Mcl-1

As noted above, BH3 mimetics displace pro-apoptotic proteins from complexes with anti-apoptotic ones, thus enhancing concentrations of free BH3-only and pore-forming members of the Bcl-2 family and promoting apoptosis. Mcl-1 interacts with various pro-apoptotic partners, including Bak, Bax, Bim, Noxa, and Puma [11]. Among them, according to several studies, the Bak protein is characterized by the highest binding affinity for Mcl-1 [11,42]. Hence, the various efficacies of BH3 mimetics A1210477 and S63845 at the used concentrations could be due to their different abilities to disrupt the complexes between Mcl-1 and Bak.

To test this hypothesis, siRNA-mediated knockdown of BAK was performed. Additionally, we downregulated other pro-apoptotic members of the Bcl-2 family, Bax (this protein, along with Bak, is one of two main effector proteins) and Bim (this protein interacts with a broad spectrum of anti-apoptotic Bcl-2 family proteins and, in contrast to Noxa and Puma, is not transcriptionally-induced by p53 in response to stress conditions) [3]. Using western blot analysis, it was shown that downregulation of Bak in HeLa cells substantially reduced PARP cleavage in cells treated with 3 μM S63845 (Figure 2A). These results were consistent with a significantly higher survival of cells treated with 1 μM or 3 μM S63845 in BAK knockdown cells, as assessed by flow cytometry (Figure 2B). In striking contrast, BAK knockdown did not influence the activity of A1210477. These data demonstrate that limited pro-apoptotic activity of A1210477 at concentration 10 μM might be due to the inefficient disruption of Mcl-1/Bak complex. Silencing of Bax or Bim did not result in changes in the survival of cells treated with either A1210477 or S63845, as assessed by western blot and FACS-analysis (Figure 2A,B).

To exclude the off-target effect of siRNA-mediated knockdown, we performed western blot analysis of cells treated with 3 μM S63845 after knockdown of BAK, BAX, or BCL2L11 (encodes for Bim) by two various siRNAs. The results were consistent for both siRNAs, thus confirming the obtained data (Appendix A). Hence, Bak represents an important factor of S63845 activity in HeLa cells.

Next, we checked whether the effects of the knockdown of various pro-apoptotic proteins on the response to S63845 were specific for HeLa cells. To this end, we performed experiments with the NSCLC line H23 with downregulated levels of Bak, Bax, or Bim, which demonstrated sensitivity to Mcl-1 inhibition. In contrast to HeLa, these cells demonstrated similar sensitivity to 1 μM and 3 μM S63845 (Figure 2C,D). Therefore, 1 μM as a maximum concentration of S63845 for experiments with H23 cell line was used. As well as in HeLa cells, both western blot (Figure 2E) and flow cytometry (Figure 2F) demonstrated that the downregulation of Bak, but not Bax or Bim, resulted in a decrease of cell death induced by BH3 mimetic S63845. We also tested whether this effect was specific for Mcl-1 inhibition, since Bak might have a more crucial role compared to Bax and Bim in the demise of HeLa and H23 cell lines independently of proapoptotic stimuli. To this end, the cytotoxic activity of cisplatin in these cell lines with knockdown of Bak, Bax, or Bim was evaluated. According to WB analysis and flow cytometry, the cytotoxic activity of cisplatin was not decreased in any of the knockdowns, both in HeLa (Figure 2G,H) and H23 (Figure 2I,J) cell lines. These data demonstrated the specificity of Bak for cell death induced by Mcl-1 inhibition. We also performed experiments with S63845 and cisplatin in BAK knockout HeLa and H23 cell lines. Consistent with siRNA-mediated downregulation of Bak, BAK knockout significantly affected cell death induced by S63845 in both cell lines. BAK knockout did not influence the response of HeLa cells to cisplatin (Appendix A). In H23 cells, BAK knockout decreased to some degree cell death induced by cisplatin. This effect might relate, at least partially, to Mcl-1, since 25 µM cisplatin caused downregulation of Mcl-1 in H23 cells in contrast to HeLa cells (Appendix A). Altogether, these data indicate that Bak is one of a crucial pro-apoptotic protein of the Bcl-2 family associated with the efficiency of BH3 mimetics to Mcl-1.

### 3.3. Bcl-xL Contributes to Resistance to Mcl-1 Inhibition

In this study, we investigated possible mechanisms of the acquired resistance of tumor cells to Mcl-1 inhibition. To accomplish this, a stable cell line characterized by the increased resistance to S63845 (HeLa S^res^) was derived. HeLa S^res^ cells were obtained by the cultivation of wild-type HeLa cells with gradually increasing doses of S63845 (up to 3 μM) (Appendix A). To exclude the possible effects of the increased drug efflux, S63845 was added to the HeLa cells in combination with the P-glycoprotein inhibitor verapamil (50 μM).

The efficiency of cell death induction by S63845 (1 μM or 3 μM) and/or cisplatin (25 μM) was then assessed. Of note, HeLa S^res^ cells were still sensitive to Mcl-1 inhibition: PARP cleavage was observed after 24 h of incubation with S63845 (Figure 3A), while the fraction of apoptotic cells was ~20% and ~30% for 1 μM and 3 μM S63845, respectively (Figure 3B). Nevertheless, HeLa S^res^ cells were significantly less sensitive to apoptosis induced by S63845 alone, as assessed by western blot (Figure 3A) and FACS-analysis (Figure 3B). Both HeLa and HeLa S^res^ cells demonstrated comparable sensitivity to cisplatin. Additionally, the combination of cisplatin with S63845 demonstrated high cytotoxic activity against parental, as well as HeLa S^res^ cells (Figure 3A,B). These data demonstrate that DNA damage can be effective in resensitizing cells to Mcl-1 inhibition. Intriguingly, western blot demonstrated that HeLa S^res^ cells expressed higher levels of anti-apoptotic protein Bcl-xL, but not of Bcl-2 or Mcl-1 (Figure 3A,C), which might represent a mechanism of acquired resistance to S63845.

To expand these data, we analyzed the impact of expression of various pro- and anti-apoptotic proteins of the Bcl-2 family on the response to S63845. For this purpose, we tested this compound in a panel of cancer cell lines of various origins (HeLa, HeLa S^res^, colon carcinoma HCT-116, lung adenocarcinoma A549 and H23, ovarian carcinoma Caov-4, neuroblastoma SK-N-BE(2) and SH-SY5Y). Three cell lines, H23, HeLa, and SK-N-BE(2), demonstrated responses to S63845 according to PARP cleavage assay, while HCT-116, A549, HeLa S^res^, Caov-4, and SH-SY5Y cells were relatively less sensitive to Mcl-1 inhibition. These results were consistent with the data of the MTS cell viability assay (Appendix A). IC50 for HeLa S^res^ was higher than for HeLa cells and was similar to other cell lines which possess low sensitivity to S63845. These data additionally confirmed resistance to Mcl-1 inhibition of the selected cell lines (Appendix A). Notably, even in H23 and HeLa cells, IC50 values were significantly higher than those reported for hematological cancer cell lines [26], which is in line with in general lower susceptibility of solid cancer cells to apoptosis induction by BH3 mimetics as compared to hematopoietic cells [24]. The levels of Mcl-1 did not strongly correlate with the sensitivity. Importantly, S63845-refractory cell lines demonstrated higher expression of Bcl-xL compared to S63845-sensitive cells (Figure 3C). The only outlier among the tested cell lines were SH-SY5Y cells, in which Bcl-xL expression was comparable to that of S63845-sensitive cells. SH-SY5Y cells highly expressed Bcl-2 and demonstrated significantly lower levels of Bak compared to another neuroblastoma cell line, SK-N-BE(2), which might explain their resistance to S63845. Taken together, these data suggest that Bcl-xL expression is an important, but not the only, factor of resistance to Mcl-1 inhibition.

### 3.4. Co-Targeting Bcl-xL and Mcl-1 Exhibits Extreme Cytotoxicity

The combination of various BH3 mimetics represents a promising strategy to increase cytotoxicity, as well as overcome resistance to individual BH3 mimetics [24]. Hence, we tested whether BH3 mimetics to Bcl-2 or Bcl-xL can potentiate the activity of Mcl-1 inhibitors and help to overcome the resistance to them. First, we analyzed the efficacy of the combination of S63845 with a selective inhibitor of Bcl-2 venetoclax on H23 and H23 S^res^ cell lines. H23 S^res^ cell line was obtained similarly to HeLa S^res^ cells with the only exception that S63845 was used at doses up to 1 μM for the generation of these cells. IC50 for H23 S^res^ increased significantly that corroborate efficient selection of resistant cell line (Appendix A). S63845 alone was used at 1 μM, while for combination with venetoclax, S63845 was utilized at 0.5 μM. Similar to the results on HeLa cells, H23 S^res^ were sensitive to S63845 alone, but to a significantly lesser extent than the parental cells, as shown by both western blot (Figure 4A) and flow cytometry (Figure 4B). The combination of 0.5 μM S63845 and venetoclax was as effective as S63845 at a higher concentration (1 μM) in H23 cells. At the same time, this combination significantly increased cytotoxicity in H23 S^res^ cells. The efficiency of S63845 + venetoclax was similar in both H23 and H23 S^res^ cells, indicating that concomitant inhibition of Bcl-2 may overcome resistance to Mcl-1 inhibition (Figure 4A,B). Similarly, venetoclax in combination with S63845 led to enhanced activation of apoptosis in HeLa S^res^ cells (Appendix A).

Furthermore, we analyzed the cytotoxicity of S63845 in combination with A1331852, which is a selective inhibitor of Bcl-xL. This combination was compared with S63845 + venetoclax on seven cell lines, including Caov-4, A549, and SH-SY5Y, as well as parental and S^res^ HeLa and H23 cells. Strikingly, the combination of A1331852 and S63845 (both 100 nM) was enough to cause extreme cytotoxicity in all tested cell lines, except SH-SY5Y cells (Figure 5A–G, Appendix A). As noted above, SH-SY5Y cells are characterized by high levels of Bcl-2, which might explain their low sensitivity to the co-targeting of Bcl-xL and Mcl-1, as compared to the S63845 + venetoclax combination (Figure 5G). At the same time, the profound processing of caspase-3 and almost full PARP cleavage were already detected after 6 h of incubation with A1331852 + S63845 in other tested cell lines (Figure 5A–F). This cytotoxic activity was not due to A1331852 alone because this compound was inefficient when used alone, except for Caov-4 cells (Figure 5F). Consistent with the data above, the S63845 + venetoclax combination induced apoptosis in H23, HeLa, H23 S^res^ and HeLa S^res^, but the activity of this combination was quite low when compared to A1331852 + S63845 (Figure 5A–D). In general, the co-inhibition of Bcl-xL and Mcl-1 already caused dramatic cell death in the early hours of incubation.

### 3.5. BAK Knockout Does Not Rescue Cells from the Co-Inhibition of Bcl-xL and Mcl-1

Since Bcl-xL and Mcl-1 are key binding partners for Bak, we hypothesized that Bak-dependent apoptosis could be the main mechanism of the pronounced proapoptotic activity of A1331852 and S63845. To test this hypothesis, we generated BAK-knockout HeLa and H23 cells using the CRISPR/Cas9 gene editing system and tested the cytotoxicity of the combination of both compounds in these cells, as well as in control cells (HeLa^control^ and H23^control^; see Material and Methods section). Since HeLa BAK-knockout (HeLa BAK-ko) cells, in contrast to H23 BAK-knockout cells (H23 BAK-ko), were derived from single clones, and for greater reliability of the results, we analyzed two different populations obtained from independent clones (designated as clone #1 and clone #2). Cell death was assessed by caspase-3 processing and PARP proteolysis. Additionally, flow cytometry was used for the measurement of cell death in HeLa cells.

The analysis showed that BAK knockout did not rescue HeLa and H23 cells from the cytotoxic effects of the combination of A1331852 and S63845. Indeed, caspase-3 processing and PARP cleavage were already intense in both control and BAK-ko cells after 4 h of incubation (Figure 6A,C). Consistent with the western blot results, flow cytometry demonstrated a high percentage of cell death in HeLa cells (Figure 6B). Thus, co-targeting Bcl-xL and Mcl-1 exhibited high cytotoxicity, even in the absence of Bak.

Additionally, the cytotoxic activity of the combination of A1331852 and S63845 in BAK-knockout HeLa and H23 cell lines with concomitant downregulation of Bax was tested. There was clear decrease in both caspase-3 processing and PARP cleavage in HeLa cells (Appendix A), as well as lower rate of cell death measured by annexin V/PI assay (Appendix A). In H23 cells, the effect of Bax downregulation on caspase-3 processing was less prominent in comparison to HeLa cell line (Appendix A). However, BAK knockout with concomitant Bax downregulation did not completely reverse cell death induced by A1331852 and S63845. It seems that even low amounts of Bax can be sufficient for effective MOMP in the case of Bcl-xL and Mcl-1 co-inhibition, especially in H23 cells where the effect of Bax knockdown was minimal.

### 3.6. Expession Profile of Bcl-2 Family Proteins Is Prognostic in Lung Adenocarcinoma

To estimate the possible utility of BH3 mimetics targeting Mcl-1 for the treatment of solid cancers, we performed the analysis of mRNA expression and protein levels in lung adenocarcinoma tissues. For this purpose, we analyzed the clinical samples and open databases. Based on the TCGA dataset (LUAD, TCGA, The Cancer Genome Atlas, https://portal.gdc.cancer.gov/, accessed on 1 September 2021), mRNA expression z-scores relative to normal samples were used to describe the changes in the Bcl-2 family proteins in cancer tissues. Our results demonstrated that the median mRNA expression of anti-apoptotic members (Bcl-2, Mcl-1, Bcl-xL) was lower in cancer than normal tissues, while the expression of pro-apoptotic members (Bid, Bim, Bax, Bak) prevailed in cancer tissues, compared to normal tissues (Figure 7A). To confirm the mRNA expression data, western blot analysis (Figure 7B) of the Bcl-2 family proteins in the clinical samples from lung adenocarcinomas and adjacent non-tumorous tissues was conducted and the log2 tumor/normal ratio of each cancer sample was calculated. As shown in Figure 7C, the levels of pro-apoptotic proteins (Bak, Bax, Bid, Bim) were increased in cancer tissues. Conversely, the levels of anti-apoptotic proteins were decreased, with Bcl-xL being the most downregulated protein among the studied ones. These data are consistent with findings from mRNA expression analyses. Such expression patterns are similar to those in S63845-sensitive cell lines, and thus suggest that Mcl-1 inhibition might be the rational in lung adenocarcinoma patients.

Finally, we evaluated the survival rate of patients with lung adenocarcinoma from the TCGA dataset, depending on the expression levels of Bak, Bcl-xL and Mcl-1. Four subsets of patients with high/low Bak and high/low Bcl-xL and Mcl-1 levels were assessed, in which high and low groups were formed based on the median of z-score mRNA expression levels, relative to normal samples (Appendix A). Of note, among these groups, patients with high Bak and low Bcl-xL demonstrated significantly different survival dependent on Mcl-1 expression (Figure 7D). Thus, concomitant high expression of Mcl-1 was associated with worse outcome, while low Mcl-1 expression demonstrated favorable prognosis compared to all other subgroups of patients. Hence, expression profile of Bcl-2 family proteins is prognostic in lung adenocarcinoma patients. Although the precise mechanism of this phenomenon is not clear, these data suggest that the Bak/Bcl-xL/Mcl-1 axis might play an important role in the response of lung adenocarcinoma to treatment and risk stratification of patients.

## 4. Discussion

Nowadays, precision medicine approaches are being actively developed, and BH3 mimetics represent a promising tool in the anticancer armamentarium of physicians. The mechanism of action of these drugs is based on the direct induction of apoptosis, which makes them extremely attractive options for the elimination of cancer cells. However, in clinical practice, BH3 mimetics are not widely used, and their application is limited to the use of venetoclax in CLL and of the combination of venetoclax with chemotherapy for the treatment of AML [24]. Further expansion of the use of BH3 mimetics might be possible due to the search for rational combinations, as well as the subgroups of patients who will benefit from these drugs. An excellent example of how these aspects can be used for rational therapy design are shown in the phase 2 CAVALLI study, in which venetoclax, in combination with rituximab-cyclophosphamide-doxorubicin-vincristine-prednisone (R-CHOP), was especially effective in high-risk Bcl-2 Immunohistochemistry (IHC) positive subgroups of diffuse large B-cell lymphoma (DLBCL) patients [43].

BH3 mimetics to Mcl-1 have only recently entered clinical trials. The preliminary results of the efficiency of Mcl-1-specific BH3 mimetic AMG-176 demonstrated moderate activity in 26 patients with relapsed or refractory multiple myeloma, in which 11 patients had stable disease as best overall response and 22 patients discontinued treatment due to progressive disease [44]. Meanwhile, AMG-176 demonstrated pronounced cytotoxicity in multiple myeloma cells in vitro [28]. Once again, these data demonstrate the high relevance of the search of biomarkers for BH3 mimetics.

In this study, we evaluated the efficacy of two BH3 mimetics to Mcl-1, A1210477 and S63845. Although both compounds bound to their target, as evidenced by the accumulation of Mcl-1, S63845 showed much higher efficacy at on-targeted doses. Mechanistic analysis demonstrated that the higher efficacy of S63845 was dependent on the pro-apoptotic Bcl-2 family member Bak, while other pro-apoptotic proteins of the Bcl-2 family, Bax and Bim, did not affect efficiency of either compound. Additionally, we detected an increased expression of the anti-apoptotic protein Bcl-xL in HeLa cells with acquired resistance to S63845. In a panel of eight solid cancer cell lines, S63845 sensitive lines demonstrated reduced Bcl-xL expression. These data indicate the key role of this protein in the resistance of cells to Mcl-1 inhibition. Importantly, tumor tissues from patients with lung adenocarcinoma demonstrated the same pattern of protein level compared to normal tissues, namely, decreased Bcl-xL and increased Bak levels.

Our findings are consistent with the data from other studies concerning the possible biomarkers of efficiency for BH3 mimetics. A number of studies have shown that not the level of the target, but rather the levels of other members of the Bcl-2 family, or even the balance between them determines the sensitivity of cells to BH3 mimetics. Thus, the significance of Mcl-1 for the acquired resistance to venetoclax and navitoclax (Bcl-2 and Bcl-xL inhibitor) has been shown [20,45]. Moreover, in clinical trials of venetoclax, the ratios of Bcl-2/Mcl-1 and Bcl-2/Bcl-xL transcripts correlated to a greater degree than the level of Bcl-2 mRNA alone with efficiency of venetoclax in multiple myeloma patients [46]. As for Mcl-1 inhibition, on a broad panel of cell lines, *BAK* mRNA expression was associated with a response to AMG-176, while mRNA expression of *BCL2L1* (encoding for Bcl-xL) was shown to be a major factor of resistance to this BH3 mimetic [28]. Similar results were obtained for S63845: in a panel of hematological cell lines, its efficiency was inversely correlated with the level of Bcl-xL mRNA, but not of Mcl-1 mRNA [26]. In our work, we show that these patterns of sensitivity or resistance to Mcl-1 inhibition are reproduced in solid tumor cell lines.

The fact that Bak and Bcl-xL determined cytotoxicity of Mcl-1 inhibition proposed that there is a tripartite interlink between these three proteins. This assumption was partially confirmed by the extreme cytotoxicity of co-inhibition of Mcl-1 and Bcl-xL, while concomitant targeting of Bcl-2 and Mcl-1 was less efficient in most of the studied cell lines. These data are consistent with some previous reports [47,48]. Of note, Bak, unlike other pro-apoptotic proteins, binds predominantly to Bcl-xL and Mcl-1 [11,13,14], which might underlie the impact of Bcl-xL and Bak in determining the efficiency of Mcl-1 inhibitors. Hence, we proposed that the dramatic cytotoxicity caused by co-targeting Mcl-1 and Bcl-xL might be diminished in the absence of Bak. Indeed, if both Bcl-xL and Mcl-1 are inhibited, no more proteins able to neutralize Bak may exist. However, surprisingly, cell death caused by the inhibition of Mcl-1 and Bcl-xL was not diminished in the absence of Bak. Thus, Bak is not an indispensable mediator of the cytotoxicity of Mcl-1 and Bcl-xL co-inhibition.

Of note, other combinations, such as venetoclax and S63845 or cisplatin and S63845, have also demonstrated their efficiency. The effectiveness of such combinations did not differ between the parental cell lines or cells with acquired resistance to S63845 alone. This indicates that the resistance shaped upon S63845 treatment does not involve mechanisms triggered by the action of cisplatin and venetoclax. Apparently, such mechanisms include the priming of cancer cells to apoptosis, including priming due to the activation of BH3-only proteins [22]. We are positive that further studies should focus on elucidating the profile of activation of pro-apoptotic proteins of the Bcl-2 family upon treatment with various anticancer agents, such as chemotherapeutic and targeted drugs, including BH3 mimetics. Some examples of such studies [49,50] exist, and the functional test called BH3 profiling should facilitate research of this issue [51]. The understanding of the complex circuitries of regulation and interaction of Bcl-2 family members is an important step in the expansion of the use of BH3 mimetics, especially in terms of the search for rational combinations.

Finally, an important question is whether BH3 mimetics could be used for the treatment of solid cancers. Nowadays, most clinical trials of BH3 mimetics include patients with hematological tumors, since they are much more vulnerable to the inhibition of Bcl-2 family members [24]. Moreover, in this study, the concentrations of S63845 required to induce apoptosis in solid cancer cell lines were significantly higher than those in hematological cell lines [26], which is consistent with the idea of higher priming to apoptosis of cells of hematological origin [6]. Our data demonstrated that tumor tissues from lung adenocarcinoma patients are characterized by the presence of the putative biomarkers for the response to Mcl-1 inhibition (i.e., high expression of Bak and low expression of Bcl-xL). Of note, patients with low Bcl-xL and high Bak, which presumably can benefit more from Mcl-1 inhibition, were clustered into two different prognostic groups, where concomitant high expression of Mcl-1 was associated with significantly poor outcome. These data suggest that expression profile of Bcl-2 family proteins has prognostic significance in lung adenocarcinoma patients. Despite data that Mcl-1 expression does not correlate with response to Mcl-1 inhibition [26], in lung adenocarcinoma patients Mcl-1 expression can be additional risk stratification factor which might help identifying patients with worse prognosis who would benefit more from Mcl-1 inhibition. Although the utility of BH3 mimetics as monotherapy in solid tumors, including lung adenocarcinoma, seems unlikely, there is still a space for the use of various combinations, which might be rational, at least in patients within the biomarker subgroups. Hence, the search for the most promising combinations and biomarkers of response/risk is of great significance for the treatment of patients with solid tumors.

## 5. Conclusions

In this report, we have found that S63845 caused apoptosis induction more effectively than another Mcl-1 inhibitor—A1210477, through a Bak-dependent mechanism. Next, we have demonstrated that cancer cell sensitivity correlates with low expression of Bcl-xL protein. Moreover, the acquired resistance to Mcl-1 inhibition could be associated with an increase in the level of Bcl-xL. Importantly, low Bcl-xL and high Bak levels of both mRNA and proteins were observed in tumor tissues from patients with lung adenocarcinoma. Altogether, our results have shown that Bak and Bcl-xL proteins are involved in the sensitivity/resistance of adherent cancer cell lines to Mcl-1 inhibition.

## Figures and Tables

**Figure 1 cancers-14-00181-f001:**
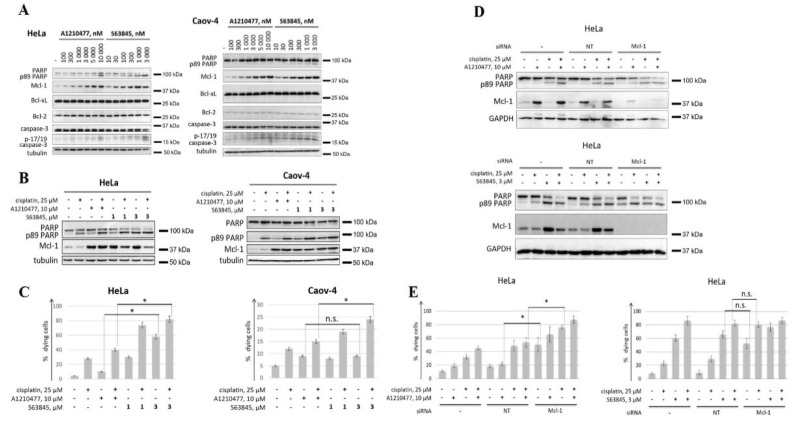
Analysis of the efficiency of BH3 mimetics to Mcl-1, A1210477 and S63845. (**A**) Western blot (WB) analysis of HeLa and Caov-4 cells upon treatment with S63845 or A1210477 at stated concentrations. (**B**,**C**) WB analysis (**B**) and FACS analysis (**C**) of HeLa and Caov-4 cells treated with 10 µM A1210477 and 1 or 3 µM S63845, either alone or in combination with 25 µM cisplatin. (**D**,**E**) WB analysis (**D**) and FACS analysis of HeLa cells treated with 10 µM A1210477 and 3 µM S63845, either alone or in combination with 25 µM cisplatin. Cells were untransfected (−), transfected with non-targeting siRNA (NT) or with siRNA to MCL1 (Mcl-1) (antisense sequence: 5′-GCATCGAACCATTAGCAGAdTdT-3′). Data from *n* = 3 biological replicates are shown as mean ± s.d., * *p* < 0.05, n.s.—not significant; % dying cells—cells positive for Annexin V-FITC and/or propidium iodide (PI). Tubulin and GAPDH were used as loading controls. Time of incubation: 24 h, time of transfection: 6 h.

**Figure 2 cancers-14-00181-f002:**
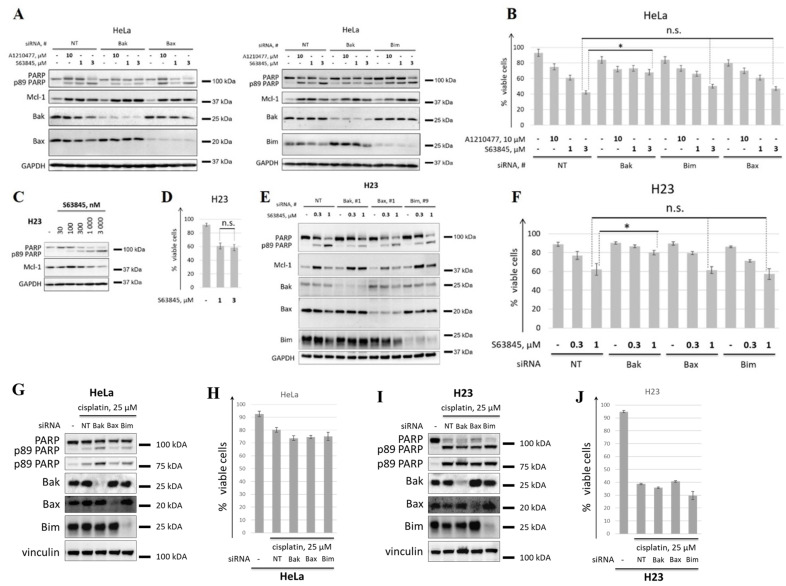
Analysis of the mechanisms underlying various efficacy of A1210477 and S63845. (**A**,**B**) WB analysis (**A**) and FACS analysis (**B**) of HeLa cells treated with 10 µM A1210477 and 1 or 3 µM S63845. Cells were transfected with non-targeting siRNA (NT) or with siRNA to BAK1 (Bak), BAX (Bax) or BCL2L11 (Bim). (**C**,**D**) WB analysis (**C**) and FACS analysis (**D**) of H23 cells upon treatment with S63845 at stated concentrations. (**E**,**F**) WB analysis (**E**) and FACS analysis (**F**) of H23 cells treated with 0.3 or 1 µM S63845. Cells were transfected with non-targeting siRNA (NT) or with siRNA to BAK1 (Bak), BAX (Bax) or BCL2L11 (Bim). (**G**,**H**) WB analysis (**G**) and FACS analysis (**H**) of HeLa cells treated with 25 µM cisplatin. Cells were transfected with non-targeting siRNA (NT) or with siRNA to BAK1 (Bak), BAX (Bax) or BCL2L11 (Bim). (**I**,**J**) WB analysis (**I**) and FACS analysis (**G**) of H23 cells upon treatment with cisplatin at 25 µM. Cells were transfected with non-targeting siRNA (NT) or with siRNA to BAK1 (Bak), BAX (Bax) or BCL2L11 (Bim). Data from *n* = 3 biological replicates are shown as mean ± s.d., * *p* < 0.05, n.s.—not significant; % viable cells—cells negative for both Annexin V-FITC and propidium iodide (PI). siRNA #1, #1 and #9 (see Appendix A. Antisense sequences: 5′-AUUUGAAGAAUCUUCGuACTsT-3′; 5′-UGAAGUUGCCGUcAGAAAATsT-3′; 5′-GCUGcAAUUGUCuACCUUCUCGGUcACTsT-3′; uppercase letters: ribonucleotides, lowercase letters: 2′-O-Methyl nucleotides, s: phosphorothioate) were used for knockdown of BAK1, BAX, and BCL2L11, respectively. GAPDH or vinculin were used as loading controls. Time of incubation: 24 h, time of transfection: 24 h.

**Figure 3 cancers-14-00181-f003:**
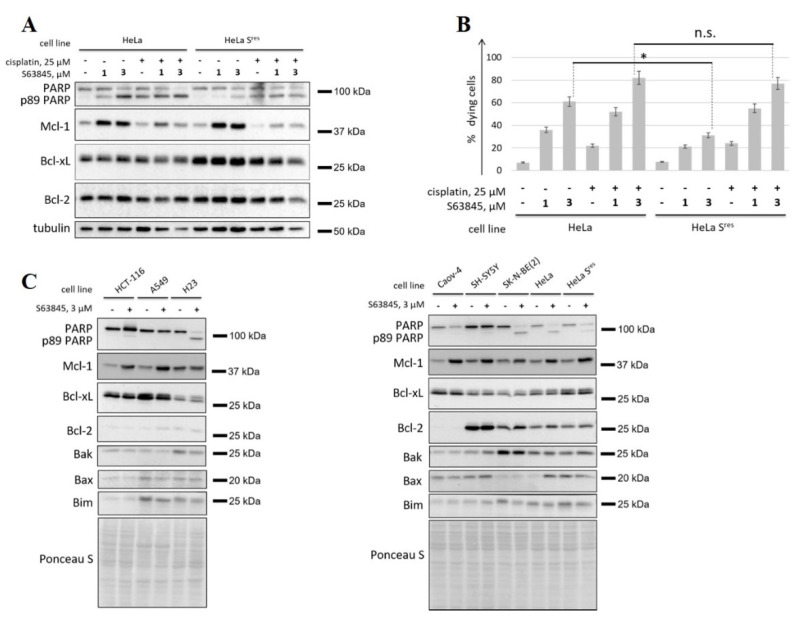
Investigation of the mechanisms of resistance to S63845. (**A**,**B**) WB analysis (**A**) and FACS analysis (**B**) of parental HeLa cells and HeLa cells with acquired resistance to S63845 (HeLa S^res^) treated with 1 or 3 µM S63845, either alone or in combination with 25 µM cisplatin. (**C**) WB of 8 cell lines (HCT-116, A549, H23, Caov-4, SH-SY5Y, SK-N-BE(2), HeLa, and HeLa S^res^) with or without treatment with 3 µM S63845. Data from *n* = 3 biological replicates are shown as mean ± s.d., * *p* < 0.05, n.s.—not significant; % dying cells—cells positive for Annexin V-FITC and/or propidium iodide (PI). Tubulin and Ponceau S were used as loading controls. Time of incubation: 24 h.

**Figure 4 cancers-14-00181-f004:**
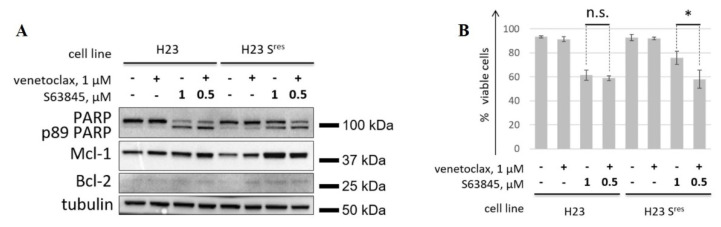
Assessment of the efficiency of concomitant inhibition of Mcl-1 and Bcl-2 in parental H23 cells and H23 cells with acquired resistance to S63845 (H23 S^res^). WB analysis (**A**) and FACS analysis (**B**) of H23 and H23 S^res^ cells treated with S63845 and venetoclax alone (1 µM each) or with their combination (0.5 µM S63845 and 1 µM venetoclax). It is important to note that the combination venetoclax 1 μM + 0.5 μM S63845 allowed to reduce by half the dose of the latter in order to prevent the development of acquired resistance to S63845. Data from *n* = 3 biological replicates are shown as mean ± s.d., * *p* < 0.05, n.s.—not significant; % viable cells—cells negative for both Annexin V-FITC and propidium iodide (PI). Tubulin was used as a loading control. Time of incubation: 24 h.

**Figure 5 cancers-14-00181-f005:**
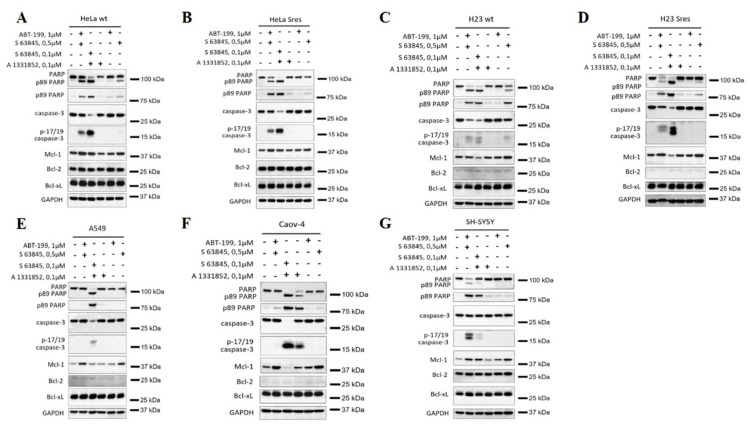
Analysis of the efficiency of combination of S63845 with BH3 mimetics to Bcl-2 (ABT-199, venetoclax) or Bcl-xL (A1331852). WB analysis of Hela (**A**), HeLa S^res^ (**B**), H23 (**C**), H23 S^res^ (**D**), A549 (**E**), Caov-4 (**F**), and SH-SY5Y (**G**) cell lines treated with S63845, venetoclax (ABT-199) or A1331852 at stated concentrations. Data from *n* = 3 biological replicates. GAPDH was used as a loading control. Time of incubation: 6 h.

**Figure 6 cancers-14-00181-f006:**
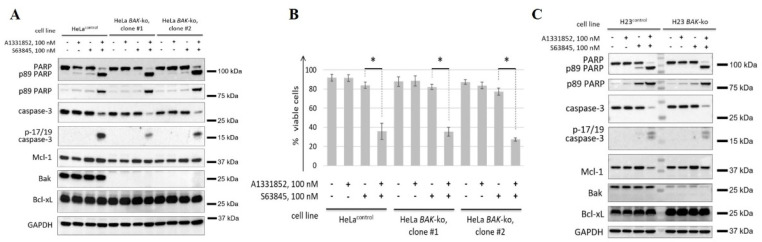
Analysis of the efficiency of concomitant inhibition of Mcl-1 and Bcl-xL in Bak-knockout cells. (**A**,**C**) WB analysis of HeLa (**A**) or H23 (**C**) cells with or without knockout of BAK1. Cells were treated with 100 nM S63845, 100 nM A1331852, or both. For HeLa cells (**A**), control cells and two independent clones with CRISPR/Cas9-mediated knockout of BAK1 were analyzed. For H23 cells (**C**), control cells and total population of cells with CRISPR/Cas9-mediated knockout of BAK1 were analyzed. (**B**) FACS analysis of HeLa cells with or without knockout of BAK1. Cells were treated as described in (**A**). Data from *n* = 3 biological replicates are shown as mean ± s.d., * *p* < 0.05; % viable cells—cells negative for both Annexin V-FITC and propidium iodide (PI). GAPDH was used as a loading control. Time of incubation: 6 h.

**Figure 7 cancers-14-00181-f007:**
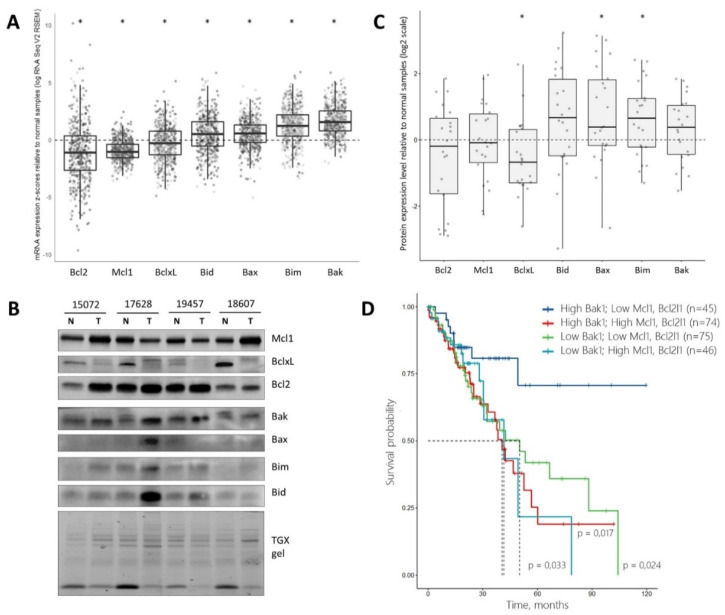
Analysis of Bcl-2 family genes mRNA expression and protein levels in lung adenocarcinoma tissues and the role in patient survival. (**A**) The mRNA expression z-scores in cancer samples relative to normal (log RNA Seq V2 RSEM) based on TCGA dataset (LUAD, TCGA, PanCancer Atlas, *n* = 510). (**B**) Representative Western blot images showing Bcl-2 family protein levels in clinical samples. Notes: T—lung adenocarcinoma tissues, N—adjacent non-tumorous tissues. TGX stain-free total protein staining is used as a loading control, * *p*-value < 0.05. (**C**) Densitometry analysis of Bcl-2 family protein levels in tumorous tissues relative to adjacent non-tumorous samples normalized on total protein level. Notes: log2 (TGX normalized protein level in tumor/TGX normalized protein level in adjacent non-tumorous sample ratio), * *p*-value < 0.05. (**D**) Kaplan–Meier survival curves in four groups with high/low expression level of Bcl2 family genes. The log-rank test with the Holm–Bonferroni correction was used to compare High Bak1/Low Mcl1/Low Bcl2l1 group of patients with other groups.

## Data Availability

The data presented in this study are available in article and Appendix A.

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
