# Peer review of "Bak and Bcl-xL Participate in Regulating Sensitivity of Solid Tumor Derived Cell Lines to Mcl-1 Inhibitors"

_cancers, 2021, doi:10.3390/cancers14010181_

Round 1
Reviewer 1 Report
In their paper the authors investigate BH3 mimetics A1210477 and S63845 in their ability to induce apoptosis in a panel of solid tumor cell lines. Combinations of the compounds with cisplatin are investigated and compared with proapoptotic activity of Mcl-1 inhibition by siRNA.
siRNA knockdown of BAK seemed to confer resistance to S63845 but not A1210477. The authors generated cell lines with increased resistance to S63845 and showed that these cells express higher levels of the anti-apoptotic protein Bcl-xL, but are still sensitive to the combination of S63845 + cisplatin. Furthermore, the combination of S63845 + a Bcl-xL inhibitor was explored. BAK knockdown did not rescue cells from the combined inhibition of Mcl-1 and Bcl-xL. The authors furthermore investigate expression of Bcl-2 family members in clinical lung cancer samples
Novelty however of findings is limited. Several papers have investigated Mcl-1inhibitors in solid tumors and hematological malignancies. A recent paper has described the combination of pharmacological Mcl-1 inhibition + BclxL inhibition in mesothelioma, including data in a xenograft mouse model. (Arulananda et al., Bcl-xL is an actionable target for the treatment of malignant pleural mesothelioma. Cell Death Discovery 2020) .
The authors use a variety of tumor cell lines, of various origins, thus making relevance of findings for a particular solid tumor entity unclear. Bcl-2 family member expression data on lung cancer samples do not substitute for this deficit.
Data on primary (clinical) tumor samples would be interesting (more easily obtained from hematological malignancies, e.g.: Tron et al., Discovery of Mcl-1 specific inhibitor AZD5991 and preclinical activity in multiple myeloma and acute myeloid leukemia. Nature Comm. 2018). However the paper is lacking such data, also no in vivo data are provided.
Other points:
The concentration of S63845 the authors use seems too high. Cell lines have been considered resistant if IC50 is < 1 µM, see Kotschy et al., The MCL1 inhibitor S63845 is tolerable and effective in diverse cancer models. Nature. 2016 Oct 27;538(7626):477-482
Figure 2 , a BAX/BAK double knockout should be included.
Figure 4a, bcl-2 Western blot, bands not clearly discernible.
Figure 5, apoptosis data are lacking.
Figure 6, a BAX/BAK double knockout should be included.
Author Response
Pleae, see the attachment

Reviewer 2 Report
This paper aimed at identifying new cell markers potentially exploitable for the design of better combination therapies in order to overcome clinical resistance to Mcl-1 inhibitors. An extensive set of well designed wb and flow cytometry experiments has been run in an attempt to reach this goal. This study provide interesting insights for future research.
In order to suitably support your claims I would recommend the following:
1) Title would be more appropriate if something like "Bak and Bcl-xL participate in regulating sensitivity of solid-tumor derived cell lines to Mcl-1 inhibitors"
2) page 1 line 35: replace "important factors of" with "co-factors for"
3) Figure 2/section 3.2:
-Authors should add general cytotox drug (e.g. cisplatin or others independent of BAK) in the flow cytometry analysis (for fig 2B and 2F)...BAK could be a factor important for Hela and H23 demise independently of the Mcl-1 inhibitor's on target effect.
-Since HELA BAK-KO cells have been developed (section 3.5) please confirm the importance of BAK in this cell model (KO vs wt; general cytox drug vs Mcl-1 inhibitors 1 and 3 uM for S63845)
3) Authors should soften claim on section 3.3 p9 with something like: "Bcl-XL contributes to " instead of "causes"
4) To expand section 3.3 claim to other cell lines, apoptosis measurements should be more quantitative. It would be needed (next to wb Fig 3C) to compare cpd IC50s obtained by dose response analyses in flow cytometry for all 8 cell lines.
5) In a similar way, IC50 values should be included to really determine whether a cell line has successfully been made resistant to a cpd (flow cytometry dose response curves Hela wt vs resistant and H23 wt vs res).
6) Figure 5: To render measurable the Mcl-1/Bcl-XL inhibitor combinatory effect, data needs to be shown in a dose-response fashion (IC50/Flow cytometry) at least for H23 Sres.
7) Section 3.6 page 13 results confusing to me (and perhaps to the prospective reader). If I understood well, the idea is to use Bcl-Xl (low) and BAK (high) as a biomarker to select patients potentially more sensitive to Mcl-1 inhibition.
-The blots (fig 7B) do not consistently confirm the mRNA analysis (Bcl-2 and Mcl-1 are up in 3 out 4 tumor samples when compared with non-tumor tissue)
-In 7B, 4 "representative" patient WB results are shown (out of 20 patients). Since in 7C a semi-quantitative analysis has been performed, P values and stat. significance levels are needed.
-On page 14 before discussion you are showing that patients with high Bak, low Bcl-XL/Mcl1 expression live longer. What would be the benefit of selecting these patients as "sensitive to Mcl1 treatment" if they already show a remarkable overall survival (after 10 Y around 70% of them are still alive)?
8) Conclusions page 16 line 618: please soften by saying something like "our results have shown that Bak and Bcl-xL proteins are involved in the sensitivity/resistance mechanisms of adherent cancer cell lines to Mcl-1 inhibitors"
Minor remarks
-Page 1 line 19: a comma got instead a Cyrillic-like character
-Fig 1 page 7: please indicate on figure Mcl-1 siRNA not only Mcl-1 (same for figure 2 Bax, Bak, Bim siRNA)
-Why is the analysis using different/a selection of cancer cell lines between figure 3 and 5 (only 4 out of 6 cancer lines remain in fig. 5)?
-fig. 5F: there is a text that should not be there "4-6 acob"
-Not clear what led to the choice of 100 nM dose for the combination study in figure 6 (previous synergy results?)
-Not clear why adenocarcinoma tissues were specifically chosen in the first place for the mRNA and subsequent analyses .
Author Response
Please, see the attachment

Round 2
Reviewer 1 Report
The authors have integrated suggestions and adressed comments of reviewers in the revised manuscript as far as possible. I think overall the paper has been improved substantially.
Author Response
We thank the Reviewer for suggestion to accept ur MS.
Reviewer 2 Report
Most of the points I raised have been addressed satisfactorily. Thanks to the authors for the additional work. I believe the paper only needs some minor corrections/additions before final acceptance.1) Please very briefly explain in figure 4,5 & 6's legends, without repeating yourself, the rationale for choosing the drug concentrations/ doses used in the combination studies (if necessary adding a reference).
2) In figure 7B legend, specify "Representative western blot images showing Bcl-2 family protein levels in clinical samples"
3) It is difficult to grasp the meaning of the new sentence on page 11 line 434 "IC50 for HeLa Sres was higher essentially that additionally confirmed resistance to Mcl-1 inhibition of the selected cell line (Fig. S5)"
please amend.
Author Response
Point-by-point response
Most of the points I raised have been addressed satisfactorily. Thanks to the authors for the additional work. I believe the paper only needs some minor corrections/additions before final acceptance.
Response:
Dear Sir/M-me,
Thank you for the high evaluation of our MS and the additional work that has been done by us. Based on your suggestions respective changes were introduced into the revised version. A detailed explanation is below.
Comment 1.
Please very briefly explain in figure 4,5 & 6's legends, without repeating yourself, the rationale for choosing the drug concentrations/ doses used in the combination studies (if necessary adding a reference).
Response:
Thank you for your comment. In figure 4 we have shown that the combination venetoclax 1 μM + 0.5 μM S63845 allowed reduction by half the dose of the latter in order to prevent the development of acquired resistance to S63845. This explanation was introduced into the Figure legend.
As we previously mentioned, concentrations for co-targeting Bcl-xL + Mcl-1 in figures 5-6 were taken after preliminary testing. These doses clearly show the difference in cell response between two settings: Bcl-2 + Mcl-1 and Bcl-xL + Mcl-1.
To our opinion, this description might “complicate” the informativeness of the figure legends.
Comment 2.
In figure 7B legend, specify "Representative western blot images showing Bcl-2 family protein levels in clinical samples"
Response:
Based on your suggestion, figure 7B legend has been corrected (marked in yellow in the revised text).
Comment 3.
It is difficult to grasp the meaning of the new sentence on page 11 line 434 "IC50 for HeLa Sres was higher essentially that additionally confirmed resistance to Mcl-1 inhibition of the selected cell line (Fig. S5)" please amend.
Response:
We are sorry for this mistake, which has been corrected in the revised version (marked in yellow).